

# Lidar measurements of noctilucent clouds at Rio Grande, Tierra del Fuego, Argentina

Natalie Kaifler[1], Bernd Kaifler[1], Markus Rapp[1], Guiping Liu[2], Diego Janches[2], Gerd Baumgarten[3], and Jose-Luis Hormaechea[4]

[1]Deutsches Zentrum für Luft- und Raumfahrt, Institut für Physik der Atmosphäre, Oberpfaffenhofen, Germany
[2]Heliophysics Science Division, NASA Goddard Space Flight Center, Greenbelt, MD, USA
[3]Institut für Atmosphärenphysik, Kühlungsborn, Germany
[4]Estación Astronomica Rio Grande, Facultad de Ciencias Astronomicas y Geofisicas, Universidad Nacional de La Plata & CONICET, Argentina

**Correspondence:** Natalie Kaifler (natalie.kaifler@dlr.de)

**Abstract.** Noctilucent clouds (NLC) are sensitive tracers of upper mesospheric temperature, water vapor and dynamics and thus open windows to study our atmosphere from very large to very small scales, including topics of climate, circulation, waves and turbulence. At northern hemisphere mid-latitudes, the occurrence of NLC seems to increase with time. NLC are weaker in the southern hemisphere, but no vertical soundings at southern hemisphere mid-latitudes had been available until now. We
determine the properties of NLC above a southern-hemisphere mid-latitude site at 53.8°S in southern Argentina. The Compact Rayleigh Autonomous Lidar provides high-resolution vertical lidar soundings since 2017. Noctilucent clouds are detected every summer, with the earliest (latest) detection on 29 November (29 January), in total 19 events of 33.8 h length, at an average height of 83.3 km, with a maximum brightness of $24 \times 10^{-10}/\mathrm{m/sr}$, an occurrence rate of 6 % and a maximum in the morning hours (5–7 UT, i.e. 2–4 LT). The latter coincides with a positive amplitude of the semi-diurnal tide of the meridional
wind as measured by the Southern Argentina Agile Meteor Radar. The ambient temperature above the site is on average too high to support local ice formation. We find no correlation with the solar flux; indeed, the latest season of 2023/2024 shows the most NLC detections. This leaves transport from more southerly, colder regions and potentially increasing upper mesospheric water vapor levels as a result of increasing space traffic as possible explanations for the occurrence and unexpectedly large brightness of NLC above Rio Grande.

## 1   Introduction

Noctilucent clouds (NLC) were discovered at northern-hemispheric mid-latitudes by visual observation of the horizon in twilight conditions (Backhouse, 1885; Jesse, 1885; Leslie, 1885). A hundred years later, observations with satellite instruments and ground-based lidars confirmed the peak occurrence and brightness of the ice clouds poleward of the polar circle (Olivero and Thomas, 1986; Hansen et al., 1989), where they remain invisible to ground-based observers in the polar daylight. Histor-
ical observations from the southern hemisphere are rare. There are reports by Jesse (1889) and (Fogle and Haurwitz, 1966) from Punta Arenas located at the southern tip of South America, the only landmass within the 50° to 60° latitude band of



the southern hemisphere except for small and remote islands. Satellite and lidar soundings of NLC above Antarctica revealed significantly higher altitudes, lower brightness, and lower occurrence compared to soundings from conjugate latitudes in the northern hemisphere (Gardner et al., 2001; Chu et al., 2003, 2006; Bailey et al., 2007; Lübken and Berger, 2007; Chu et al.,

2011; Hervig et al., 2013), making ground-based observations of NLC in the southern hemisphere even more challenging. In recent decades, the number of observations in the northern hemisphere appeared to increase, and efforts were made to uncover the origins of mid-latitude NLC and study their possible relation to climate change (von Cossart et al., 1996; Nielsen et al., 2011; Hultgren et al., 2011; Gerding et al., 2013a; Russell III et al., 2014; Hervig et al., 2016). Meridional transport of cooler polar air masses to mid-latitudes by means of tidal winds, and sometimes influenced by planetary wave activity, have been

identified as a possible cause (e.g. Nielsen et al., 2011; Hultgren et al., 2011). In one case of a NLC detected by lidar at 48°N in Germany, NLC particles were likely transported confined within the cold phases of a gravity wave that propagated from higher latitudes to the observation site in an upper mesospheric duct generated by a 2-day planetary wave (Kaifler et al., 2018). Another mechanism that is discussed is the direct injection of water vapour into the lower thermosphere by exhaust of orbital rockets, which can be transported within few days over large distances (Siskind et al., 2003; Stevens et al., 2005a).

The development of DLR's Compact Rayleigh Autonomous lidar (CORAL) made it possible to deploy powerful lidar systems for middle atmosphere research to locations worldwide without the need for sophisticated local infrastructure and availability of on-site operators (Kaifler and Kaifler, 2021). With the purpose of studying gravity waves within the stratospheric hotspot of South America, the CORAL system was deployed to Rio Grande, Tierra del Fuego, Argentina (53.8°S, 67.8°W, 18 m above sea level), in November 2017 (Kaifler et al., 2020). This site in the lee of the Andes is often free of tropospheric clouds

and thus provides better observation conditions than sites at the South American west coast or the Antarctic Peninsula. Prior to the deployment, no sightings of NLC north of 54°S have ever been reported from this or any other longitude in the southern hemisphere. So the detection of NLC by the CORAL instrument was rather unexpected, given that NLC observed in ground-based images are typically located 500 to 1000 km poleward of the observation site. After the first detection of NLC by CORAL, we installed ground-based cameras in 2019 for potential joint observations of NLC above Tierra del Fuego. The instruments are

described in Sec. 2. In Sec. 3.1 we present the set of NLC detections obtained with the CORAL instrument during seven summer seasons since November 2017, as well as simultaneous and additional detections of NLC in images of the ground-based cameras (Sec. 3.2). To characterize the thermal background, we analyze lidar temperature measurements in the upper mesosphere in Sec. 3.3. The NLC parameters height and brightness are discussed in Sec. 4.1, and possible mechanisms facilitating the occurrence of NLC at mid-latitudes, considering meridional winds, tides and planetary waves based on observations provided

by the co-located Southern Argentina Agile Meteor Radar (SAAMER), are explored in Sec. 4.2. We discuss influences on the start of the NLC season and intra- and interannual variations of the occurrence rate in Sec. 4.3 before concluding in Sec. 5.



## 2 Instruments

### 2.1 CORAL lidar

The CORAL lidar at Estación Astronómica Río Grande, Argentina, is primarily used to study gravity waves by means of an-
alyzing temperature perturbations in the stratosphere and mesosphere between 30 and 90 km altitude (Reichert et al., 2021).
The instrument is a high-power Rayleigh backscatter lidar with capabilities for nighttime autonomous operation, thus maxi-
mizing the observation time by taking measurements whenever weather conditions allow for operation of the instrument. In
the seven summers since 2017, CORAL measured temperature during 282 h (120 nights) in November, 231 h (120 nights) in
December and 269 h (125 nights) in January (updated from Reichert et al., 2021). The number of measurement hours per day
is significantly reduced compared to winter due to the short nights. Between 20 November and 31 January, measurements were
generally undertaken between 1:53 UT and 7:41 UT (22:53 and 4:41 LT), and within $\pm 10$ days from solstice the maximum
measurement duration per night was 5.2 h. Because the scientific focus of the instrument is on gravity waves that propagate
freely into the middle atmosphere only in winter, no daylight filters were included in the initial instrument design.

NLC are detected in lidar data by comparing the backscatter signal to a model density profile fitted to the measured profile
below NLC altitudes in a similar way as described in Kaifler et al. (2022). The NLC retrieval uses photon counts of the upper
channel of CORAL that are binned to 100 m vertical resolution. A detector dead-time correction is applied and the data are
rebinned in time to 10 s and 10 min to obtain a high- and a lower-resolution NLC dataset. To obtain an error estimate, we carry
out 200 Monte Carlo simulations where we perturb the photon counts with a random factor between 0 and 1 multiplied by their
square root (Poisson statistics). We remove a linear background estimated between 125–220 km altitude and account for range
by multiplication with the range squared. The resulting profiles that are now proportional to air density are then normalized to a
reference density profile between 45–50 km altitude. For this, the NRL-MSIS2 density profile of 10 Jan 2018 at 4 UT at $54°$ S
and $292°$ E is used. The profile has a vertical spacing of 1 km and ranges from 0 to 100 km altitude. Before the normalization,
we interpolate this reference profile to match the altitude vector of the lidar observations. The volume backscatter coefficient
$\beta$ in units of $10^{-10}$/m/sr (also termed "NLC brightness") is calculated from Eqn. 2 of Kaifler et al. (2022). The final $\beta(z)$
is calculated as the mean of the 200 Monte Carlo runs and the uncertainty $\Delta\beta(z)$ as the standard deviation. We discriminate
between the NLC layer and the background using the condition $\beta(z) > s\Delta\beta(z)$ with a threshold of $s = 2.5$. For CORAL,
$\Delta\beta(z)$ is on the order of $0.03 \times 10^{-10}$/m/sr.

The capability of the instrument for the detection of even faint NLC, as well as for measuring the temperature in the upper
mesosphere in regions without NLC, was demonstrated by Kaifler et al. (2018). A detailed description of the temperature
retrieval is given by Kaifler and Kaifler (2021). Following a pyramid structure, profiles with increasing resolution are computed
using seed temperatures from previously retrieved lower-resolution profiles. The initial seed temperature is taken from the
closest temperature profile of the Sounding of the Atmosphere using Broadband Emission Radiometry satellite instrument. In
this work, we use two CORAL temperature data sets: nightly mean profiles and hourly profiles. Both data sets have a vertical
resolution of 900 m.



## 2.2 NLC cameras

Encouraged by the lidar's initial NLC detections, we installed three automatic GoPro cameras for NLC observations in Argentina. The sites and viewing directions were chosen such that the cameras' field of views include the CORAL lidar beam and extend the observed area towards south, enabling comparisons between the vertical lidar soundings and the spatial imaging in a common volume. A further goal was the detection of NLC events at latitudes south of Rio Grande. Starting on 14 November 2019, two cameras located at Rio Gallegos at the Observatorio Atmosferico de la Patagonia Austral (51.6° S, 69.3° W, about 260 km north of Rio Grande) and Ushuaia (54.8° S, 68.3° W, 120 km south of Rio Grande) observe the sky in direction of the southern horizon. A third camera was installed in Rio Grande in December 2020.

## 2.3 Meteor radar

Co-located with the CORAL instrument in Rio Grande is the SAAMER radar (Fritts et al., 2010). SAAMER is a high-power, eight-beam meteor radar operating at 32.55 MHz. The inferred radial velocities of advected meteor trails at off-zenith angles between 15° and 50° are used to derive mean hourly zonal and meridional winds between approximately 70 km and 110 km altitude at 3 km vertical resolution. SAAMER measurements have revealed significant tidal and planetary wave amplitudes (Fritts et al., 2010). Most relevant for this work are a significant semidiurnal tide and planetary waves with periods between 2 d and 16 d.

## 3 Results

### 3.1 Lidar NLC detections

Using the CORAL instrument, NLC were detected above the site in Rio Grande in every season since the beginning of measurements in 2017. The first detection succeeded on 10 January 2018 and included a 3 h-long and unexpectedly bright NLC display. The NLC was also visible in camera images used to monitor the surroundings for tropospheric clouds. The detection by the vertically pointed lidar implies that the ice cloud must have been visible from more equatorward locations looking towards the southern horizon as well, but no sightings were reported, likely due to the lack of incidental observers in the very early morning hours. The first season ended with a second weak detection one week later, on 18 January 2018, with measured backscattered coefficients that were barely above the detection threshold. The next season started two weeks earlier than the 2017/2018 season, on 24 December 2018, with a bright two-night display, followed by two weaker displays on 4 January and 9 January 2019. The 2019/2020 season offered no bright displays but started very early on 29 Nov 2019 and ended on 31 Dec 2019. The 2020/2021 season held only one display on 3 January 2021, yet it was again of high brightness. This event is exceptional because it includes the lowest altitude of any NLC detections at Rio Grande (81.7 km). The single detection of the next season occurred very late. On 23/24 January 2022, NLC were detected right after sunset, but, due to forecasted rain, the lidar turned itself off in the middle of the night. After a manual start of the instrument at 5:30 UT, the NLC layer turned out to be very bright and at remarkably high altitudes. In the 2022/2023 season, again, a single detection happened even later



**Table 1.** List of all lidar NLC soundings above Rio Grande in the seven summer seasons of 2017/2018 to 2023/2024 based on 10 min resolution. dfs denotes days from summer solstice on 21 December. Times indicate the first and last NLC detection in this night.

| Date | dfs | Hours (UT) | Duration (h) | Altitude (km) | $\beta_{\max}(10^{-10}/\mathrm{m/sr})$ |
|------|-----|-----------|--------------|---------------|----------------------------------------|
| 10 Jan 2018 | 19 | 4:19 – 7:29 | 3.3 | $82.7 \pm 0.6$ | 18.1 |
| 18 Jan 2018 | 27 | 3:18 – 4:48 | 1.3 | $85.2 \pm 0.4$ | 0.2 |
| 24 Dec 2018 | 2 | 3:47 – 6:37 | 2.7 | $83.0 \pm 1.4$ | 3.1 |
| 25 Dec 2018 | 3 | 2:30 – 7:00 | 4.3 | $83.4 \pm 0.5$ | 7.4 |
| 4 Jan 2019 | 13 | 2:36 – 4:56 | 2.2 | $84.8 \pm 1.7$ | 2.0 |
| 9 Jan 2019 | 18 | 3:35 – 6.25 | 1.8 | $82.3 \pm 1.4$ | 0.1 |
| 29 Nov 2019 | -23 | 2:01 – 7.01 | 4.0 | $83.3 \pm 1.0$ | 0.8 |
| 26 Dec 2019 | 4 | 2:44 – 5:04 | 0.8 | $83.0 \pm 2.4$ | 0.3 |
| 29 Dec 2019 | 7 | 6:42 – 6:53 | 0.3 | $85.2 \pm 0.1$ | 0.6 |
| 31 Dec 2019 | 9 | 2:27 – 6:07 | 2.5 | $84.8 \pm 1.6$ | 0.4 |
| 3 Jan 2021 | 12 | 2:11 – 7:11 | 3.0 | $82.3 \pm 2.5$ | 7.0 |
| 24 Jan 2022 | 33 | 1:56 – 7:56 | 3.7 | $84.5 \pm 0.4$ | 10.5 |
| 29 Jan 2023 | 38 | 2:35 – 7:55 | 2.3 | $84.9 \pm 1.1$ | 0.6 |
| 26 Dec 2023 | 5 | 3:54 – 6:54 | 3.2 | $86.0 \pm 0.9$ | 1.5 |
| 27 Dec 2023 | 6 | 4:52 – 6:42 | 2.0 | $88.6 \pm 0.5$ | 0.2 |
| 30 Dec 2023 | 9 | 5:44 – 7:04 | 1.5 | $84.5 \pm 1.4$ | 0.6 |
| 15 Jan 2024 | 24 | 3:44 – 7:24 | 2.2 | $88.0 \pm 2.5$ | 0.5 |
| 18 Jan 2024 | 27 | 2:11 – 5:41 | 2.7 | $86.1 \pm 1.8$ | 1.1 |
| 27 Jan 2024 | 36 | 5:12 – 7:52 | 2.7 | $84.5 \pm 1.8$ | 3.0 |

on 29 Jan 2023, but the NLC were of low brightness. Interestingly, in 2023/2024, a total of six events of low brightness were recorded. Table 1 summarizes the key parameters of all NLC observed to date.

Fig. 1 shows the measured NLC brightness for main NLC events from 2017 to 2024. The coarse temporal resolution of 10 min results in an increased signal-to-noise ratio that facilitates the detection of weak events. Many NLC detections span multiple hours and occur in coherent layers. The NLC layer width is highly variable, from few hundred meters (e.g. Fig. 1j) to 4 km (Fig. 1i). Weaker layers emerging at high altitudes descend to altitudes around 82 km while increasing in brightness, e.g. Fig. 1a,f,g,j,p. This behaviour is rather typical and also known from lidar observations in the northern hemisphere (e.g. Nussbaumer et al., 1996). On many days sudden changes in NLC layer altitude occur (see Fig. 1a,d,f,h,i). These changes are likely due to the vertical motion of the background atmosphere induced by gravity waves. For bright NLC displays that offer a high signal-to-noise ratio, we were able to increase the temporal resolution to 10 s. The high-resolution data sets shown in



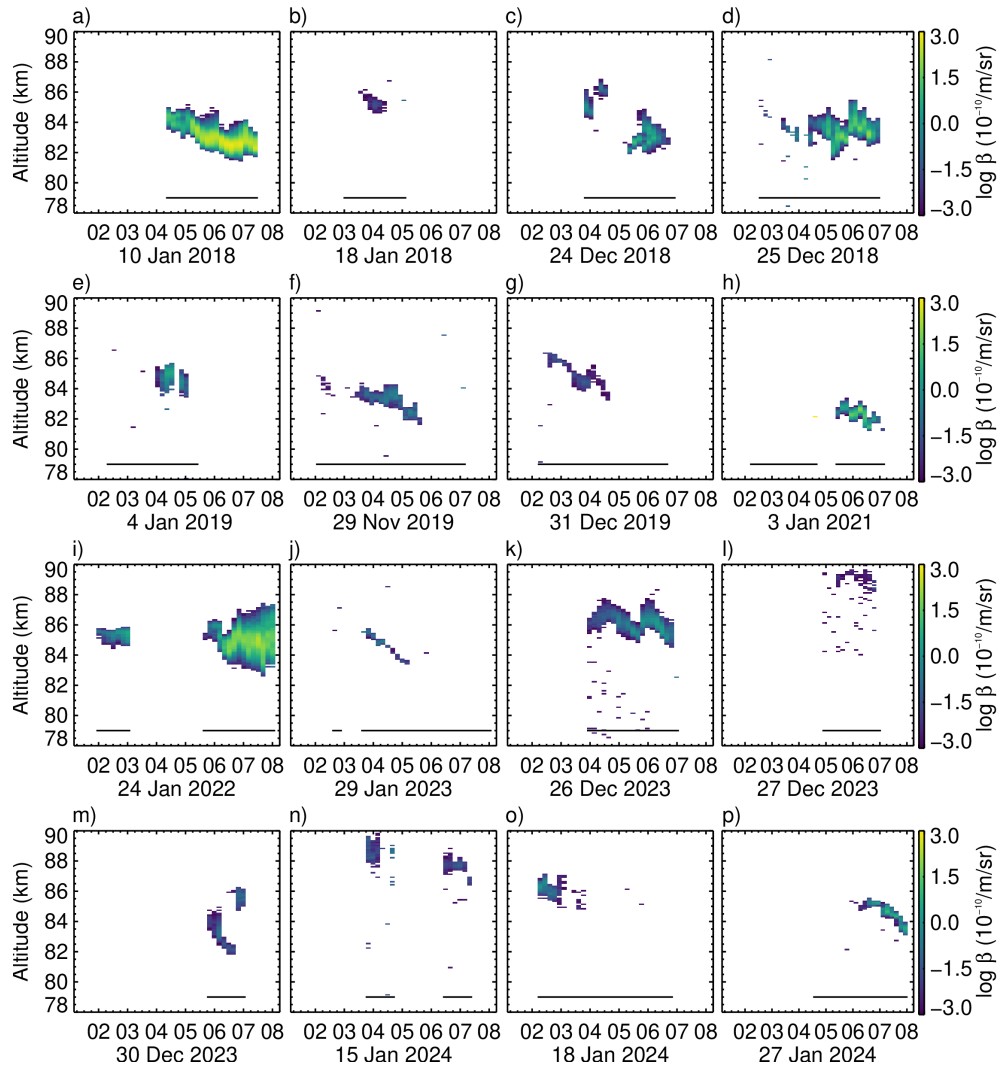

**Figure 1.** Main NLC detections at 100 m × 10 min resolution. Figure axes are identical to facilitate easy visual comparisons. The horizontal lines at 79 km altitude indicate measurement times.

Fig. 2 reveal a range of oscillations imprinted on the NLC layers by gravity waves and instabilities produced by breaking of these waves, with periods ranging from few minutes to about one hour. In particular, on 10 January 2018 between 5:20 UT to 6:50 UT (Fig. 2a), possibly until the end of measurement at 7:30 UT, the NLC layer is perturbed at periods below the Brunt-Vaisälä period. During the 30 min from 5:30 UT to 6:00 UT, modulations of the wide NLC layer with displacements of almost 2 km were likely caused by Kelvin-Helmholtz instabilities exhibiting tube and knot dynamics (Fritts et al., 2023). Some NLC layers are very thin with widths below 500 m, e.g. Fig. 1d,h. A stack of three narrowly-spaced layers was detected on 24 December 2018 shortly before 6 UT (Fig. 2b). Other layers are very wide, e.g. a layer approaching a vertical extent of





4 km was detected on January 2022 (Fig. 1i and 2f). Except for the two brightest NLC recorded on 10 January 2018 and 24 January 2022, the layers seem to have (mostly) faded by the time the lidar turned off due to sunrise, which occurs at Rio Grande
between 7:08 and 8:07 UT in the relevant period of time. The accumulation of NLC at the end of the night is discussed with respect to the effect of thermal tides in Sec. 4.2.

## 3.2 Camera observations

Coincident and common-volume observations of NLC with ground-based lidars and cameras are challenging, as only a narrow latitude band offers the right combination of background conditions for both instruments. Optimal observing conditions are
twilight for cameras and darkness for lidars. Difficulties with the interpretation of ground-based images concern mostly the foreground that obstructs the view towards the horizon and may also degrade the image quality due to unfavourable lighting conditions, e.g. by reflections from street lights in haze. Moreover, the analysis of images is often complicated by a partly overcast sky that allows for only fragments of the NLC layer to be seen. For joint observations with the cameras in Rio Gallegos and the lidar at Rio Grande, observing conditions must be good at both places. But even in cases with good observing
conditions, detections of NLC using cameras are much more likely given that the FOV of cameras are extremely large compared to lidars. In fact, NLC can be detected in camera images at locations close to the lidar beam and yet remain unvisible to the lidar. In Table 2 we list all visual detections of NLC. It is remarkable that for nights with detections of NLC by lidar, NLC were visually confirmed by at least one camera in almost all cases since the cameras were installed. In the 2019/2020 summer season, we identified a total of ten occurrences of NLC detected by a cameras, among them are five cases where the visual detection
is without doubt. For two of the events with uncertain detections, the presence of NLC was confirmed by lidar. In comparison, and also in agreement with the lidar observations, the summer season 2020/2021 offered fewer nights with NLC. 16 nights in December 2020 were without NLC detections from either Rio Gallegos or Rio Grande and the remaining nights were overcast. A first coincident observation with lidar and cameras succeeded on 3 January 2021, followed by another detection of NLC in camera images one week later. Towards the very end of the 2021/2022 season, NLC were sighted in a region east of CORAL
in the morning hours of 19 January 2022 from both Rio Grande and Rio Gallegos, followed five days later by a bright and wide display with coincident detections by lidar and cameras. In the 2023/2024 season, four coincident events were recorded. Overall, our data suggest that NLC can be observed visually from Tierra del Fuego two to ten nights per season, either after sunset or before sunrise. The total hours of visual observations in five seasons is more than 23 h.

 The coincident detections on 3 January 2021 and 24 January 2022 showed bright and structured NLC. In the morning of 3
January 2021, bright NLC covered large parts of the sky (one image is shown in Fig.3). The projected and geo-located images suggest that the area covered by NLC is at least 500 km $\times$ 800 km. Furthermore, the images show evidence of localized small-scale gravity waves of few kilometers horizontal wavelength that move in different directions and interfere with each other. The strongest activity appears at latitudes south of Tierra del Fuego. Above Rio Grande, the NLC layer, although confirmed by the lidar soundings, is too weak to show up in the twilight images, making a direct comparison between the height-resolved
backscatter signal and the imaged NLC layer impossible. Yet observations from both instruments agree qualitatively. The lidar profiling shows a periodic modulation of the NLC layer with 1 km to 1.5 km vertical displacement with maxima around





**Figure 2.** Selection of the strongest NLC displays shown at 100 m × 10 s resolution.

6:14 UT and 6:45 UT. The modulation is likely caused by gravity waves that give rise to smaller-scale variability that is imaged above the Drake passage. Evidence for the very same small-scale dynamics are also found throughout the lidar soundings in the





**Table 2.** List of NLC detections using cameras in Rio Gallegos (RIGA), Ushuaia (USHU), and Rio Grande (RIOG). An x denotes a successful detection, a '∼' a possible detection, a '-' the confirmed absence, and a blank that the instrument was not operational or that it was cloudy. In the column 'Hours' the first and last NLC detection of the night is noted. Lidar detections are noted in the last column for reference.

| Date | dfs | Hours (UT) | Duration (h) | RIGA | USHU | RIOG | Lidar |
|------|-----|-----------|--------------|------|------|------|-------|
| 5 Dec 2019 | -16 | 1:31 – 2:43 | 1.2 | x | | | - |
| 10 Dec 2019 | -11 | 2:50 – 3:50 | 1.0 | ∼ | ∼ | | - |
| 11 Dec 2019 | -10 | 2:08 – 2:35 | 0.5 | - | x | | - |
| 23 Dec 2019 | 2 | 2:40 – 3:30 | 0.8 | ∼ | ∼ | | - |
| 26 Dec 2019 | 5 | 5:10 – 5:30 | 0.3 | - | ∼ | | x |
| 29 Dec 2019 | 8 | 2:40 – 4:00 | 1.3 | - | x | | x |
| 30 Dec 2019 | 9 | 5:10 – 6:20 | 1.2 | x | x | | - |
| 31 Dec 2019 | 10 | 2:40 – 3:21 | 0.7 | | ∼ | | x |
| 6 Jan 2020 | 16 | 2:20 – 3:56 | 1.5 | ∼ | | | x |
| 18 Jan 2020 | 28 | 2:45 – 6:00 | 2.8 | x | | | - |
| 3 Jan 2021 | 13 | 5:20 – 7:19 | 2.0 | ∼ | | ∼ | x |
| 10 Jan 2021 | 20 | 6:20 – 7:10 | 0.8 | x | | | - |
| 19 Jan 2022 | 29 | 6:30 – 7:00 | 0.5 | ∼ | | ∼ | - |
| 24 Jan 2022 | 34 | 6:30 – 8:12 | 1.8 | x | | x | x |
| 26 Dec 2023 | 5 | 5:29 – 6:40 | 1.2 | | | x | x |
| 29 Dec 2023 | 8 | 4:49 – 7:10 | 2.4 | x | | x | |
| 30 Dec 2023 | 9 | 2:48 – 7:07 | | x | | x | x |
| 18 Jan 2024 | 28 | 1:52 – 3:29 | 1.6 | x | | x | x |
| 27 Jan 2024 | 37 | 7:05 – 8:22 | 1.3 | x | | x | x |

form of very short-period (around 1 min) modulations of the NLC layer with vertical displacements of few hundred meters. A
detailed analysis of the second coincident and common-volume observation on 24 January 2022 will be presented in a separate work, which will include additional OH imaging and meteor radar datasets that are available for this particular event.

### 3.3 Lidar temperature

Inherently, no temperature can be measured within NLC layers by Rayleigh lidar, as scattering by ice particles contaminates the lidar return signal, which normally, without the presence of ice particles, just comprises Rayleigh scattering from air at
these altitudes. However, for all nights without NLC (which account for most measurements), upper mesospheric temperatures can be retrieved from lidar measurements. Fig. 4a shows a smoothed composite of seven years of measurements in the summer months. Minimum temperatures of about 160 K are reached at around 88 km altitude in the beginning of January. The transition



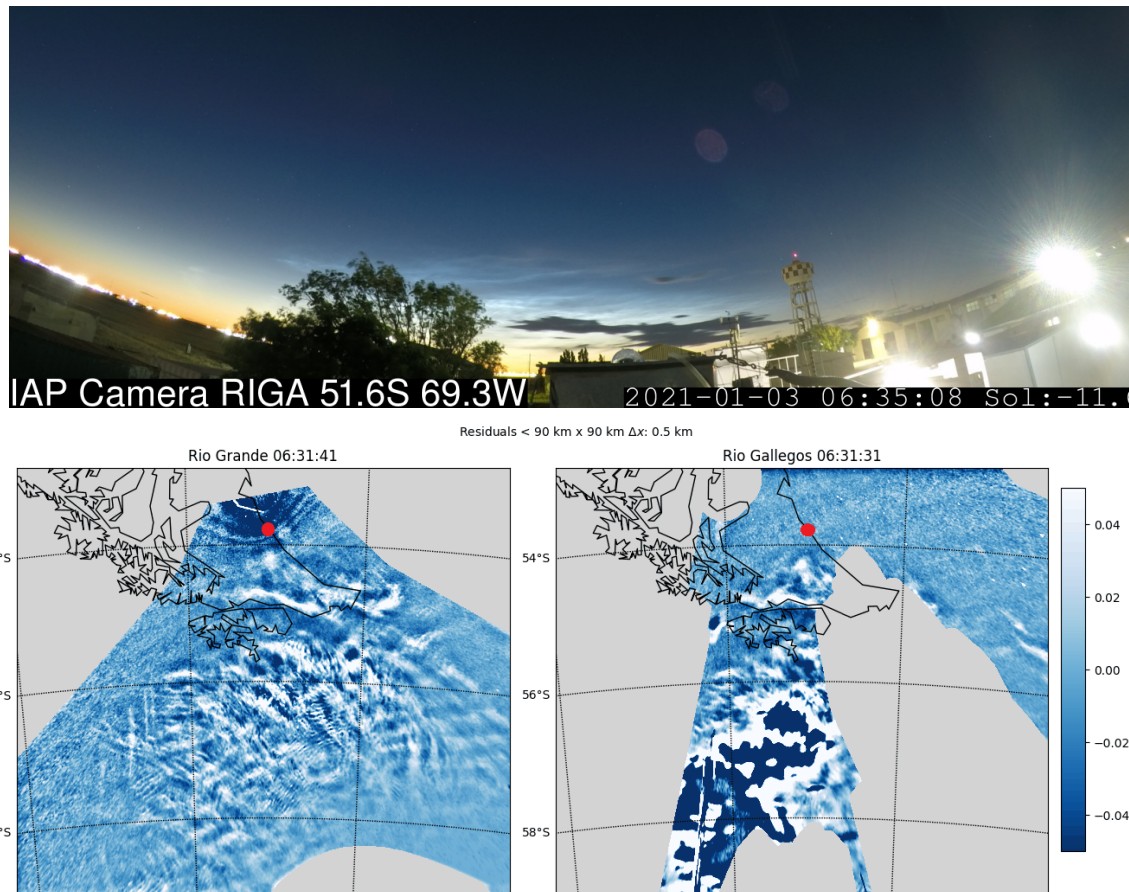

**Figure 3.** Top: Image taken at Rio Gallegos on 3 January 2021 6:35 UT. The silvery-white clouds above the horizon are NLC, while tropospheric clouds appear dark. Bottom: Two images from Rio Grande and Rio Gallegos projected to 83 km altitude on an UTM grid. The foreground has been masked, the sky background removed, and the contrast enhanced to highlight structures within NLC. The red dots mark the location of Rio Grande.



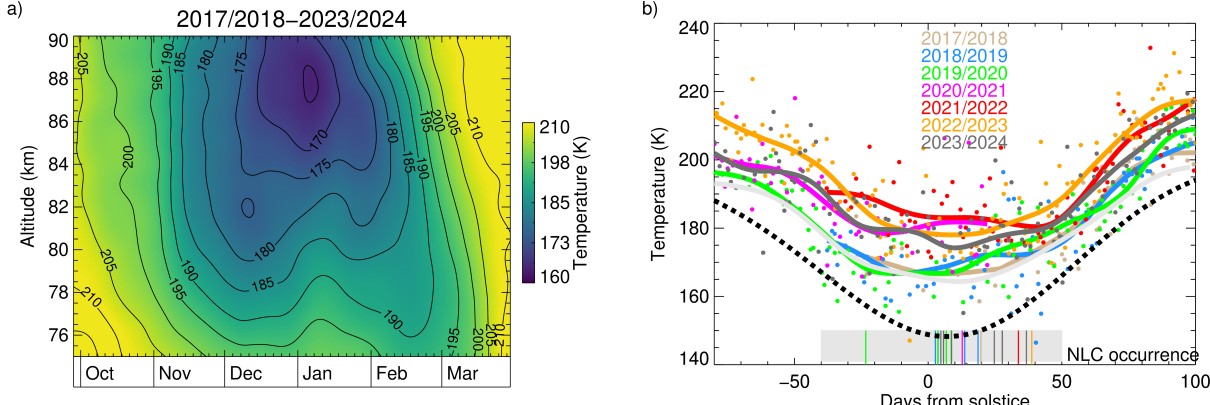

**Figure 4.** (a) Nightly mean CORAL temperatures, excluding nights with NLC, smoothed with Hanning windows of 30 d and 2 km. (b) Nightly mean CORAL temperatures averaged between 82 km and 85 km altitude, excluding nights with NLC. The time series are smoothed with a Hann filter of 50 d length (solid lines, per season). The nights with NLC detections are marked by vertical lines at the bottom. The grey line shows mean SABER temperatures of up to 800 km distance to CORAL. The black dotted line shows the mean temperature at 84 km altitude at Kühlungsborn (54°N) which is close to the conjugated point in the northern hemisphere (Gerding et al., 2008).

from winter to summer and summer to winter conditions occurs in the first half of November and in the second half of February, respectively. Fig. 4b allows a more detailed look at temperature variations within and between years. It shows all nightly mean
temperatures measured by CORAL between October and March and averaged between 82 km and 85 km altitude as colored dots. The yearly series are smoothed with a Hanning window of 50 d length (solid lines) to highlight the seasonal variation. The typical variation in temperature from one night to another (dots) is in the order of 20 K. This variability is not due to measurement errors, which are around 3 K. It should be noted that nights are short in the summer season, and taking the nightly means is unlikely to average out the effect of gravity waves. Hence, we believe that the large variability is indicative of
the upper mesosphere being disturbed by gravity waves. There is also significant variation between years, with the first three seasons exhibiting lower temperatures than the last four.

As demonstrated by Fig. 4b, NLC detections occur well within the period of time when the upper mesosphere is coldest. Although nightly mean temperatures below 160 K are occasionally measured, the mean temperature on days without NLC is generally above 170 K, and thus more than 20 K above the temperature that is required for the formation of NLC. The 2002-
2007 mean for Kühlungsborn at 54°N (dashed black line in Fig. 4b) shows temperatures much closer to and below the frost point in the middle of the season (gray box). In comparing the temperatures at both places, one would expect fewer NLC at Rio Grande and a shorter season, but neither is the case (see the discussion in Sec. 4.3). We conclude that the NLC detections are unexpected with respect to the thermal background that seems, in general, not favourable for local formation of NLC.





## 4 Discussion

### 4.1 NLC brightness and altitude

We turn to statistical analysis of the NLC parameters measured by lidar. Fig. 5a shows the mean brightness profile, i.e the sum of all profiles with significant NLC divided by the number of profiles. Colors indicate the date of the observation. NLC backscatter originates from the 81–87 km altitude region and is dominated by a number of bright events, most notably 10 January 2018 around 82.6 km altitude and 24 January 2022 around 85 km altitude. The brightness of four out of the 19 events is above $\beta = 14 \times 10^{-10}/\text{m/sr}$, while 14 events are of low brightness below $\beta = 4 \times 10^{-10}/\text{m/sr}$. A similar distribution was reported by Gerding et al. (2013b) from multi-year observations at 54°N in the northern hemisphere. The weighted mean altitude of all CORAL profiles is 83.3 km. Extrapolating the NLC altitude from South Pole, McMurdo and Rothera (Fig. 2 of Chu et al., 2011) to 54°S, one would expect a mean NLC altitude at Rio Grande of approximately 83.5 km. This is surprisingly close to our value despite our statistics being limited by the number of observations. The mean centroid altitude from Gerding et al. (2013b, 54°N) is 82.7 km and thus 600 m below our value. Gerding et al. note that strong NLC tend to occur at lower altitudes and have a smaller vertical extent compared to weak NLC. This is in contrast to our NLC observations in the southern hemisphere which can be characterized as high altitude and having a large vertical extent. But we also note that our observations are dominated by singular events, most notably the event on 24 January 2022.

### 4.2 Local time dependence

NLC were detected by lidar more often in the early morning hours before sunrise than after sunset. In Fig. 5b we show detections of NLC as function of time of day. The peak in brightness between 5–7:30 UT suggest that the occurrence of NLC is tied to the phase of the meridional wind. The analysis of radar wind data presented in Liu et al. (2020, their Fig. 1) reveals signatures of a strong semi-diurnal tide in the upper mesosphere with meridional winds exceeding 10 m/s above Rio Grande between 6–8 UT in the yearly mean. Low zonal winds and positive meridional winds at Rothera (1500 km south of Rio Grande) around 0 UT and lasting a few hours favor transport of cold air masses from more southerly regions to Rio Grande. But a steady mean flow of 10 m/s over a period of time of 8 h is not sufficient for transporting NLC particles from polar latitudes south of 60° S to Rio Grande. We therefore investigate SAAMER data for the specific dates of our NLC measurements.

We analyze hourly values of the meridional wind component at 84 km altitude that was measured by the SAAMER radar during five seasons from 2017–2022. Specifically, we look at the hourly winds at times when the NLC were brightest, and determine the mean wind 6 h, 12 h and 24 h prior to these times as done by Pokhotelov et al. (2019, their Fig. 6). Table 3 lists our results. Indeed, in quite a few cases the wind speeds were significantly larger than 10 m/s. It is obvious that the larger the wind speed, the more likely is transport from polar latitudes toward Rio Grande. The only dates where transport from the south seems unlikely based on this analysis are 9 Jan 2019, 26 Dec 2019 and 3 Jan 2021. The former two dates represent very weak NLC events.

To study the influence of planetary waves and tides on NLC in more detail, we performed a wavelet analysis of the meridional wind at 84 km using a Morlet wavelet. Spectral power for periods between 6 h and 16 d are presented in Fig. 6. The most



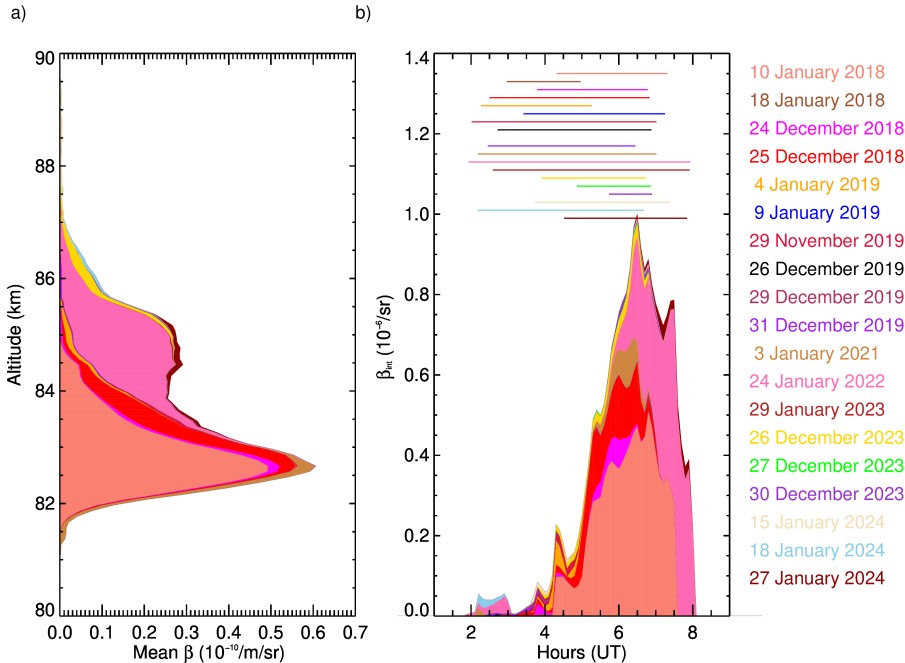

**Figure 5.** Mean brightness of all NLC events (10 min resolution data set) stacked on top of each other as a function of a) altitude and b) local time. Dates are indicated by color as shown in the legend.

prominent feature is the concentration of spectral power at periods of around 2 days, in particular in January, indicating a quasi-2 day planetary wave (Baumgaertner et al., 2008). Only one NLC observation, the short and weak detection on 4 January 2019 (Fig. 6b), coincides with significant planetary wave activity (1.6 d period). For all other NLC detections there was no

significant planetary wave activity. This result is in agreement with the finding of Siskind and McCormack (2014) that periods of time with enhanced quasi-2 day wave activity are related to high-latitude temperature enhancements that are linked to disappearance of NLC. (Merkel et al., 2003) and (Merkel et al., 2008) detected a pattern in noctilucent cloud brightness related to the 5-day planetary wave from satellite measurements. At Rio Grande in the summers of 2017–2021, the 5-day planetary wave however was weak. We conclude that the observed recurrence of NLC after 2, 3 or 5 days in some seasons happened by

chance and is not related to planetary waves. Focusing on tidal periods below 1 d, note that the majority of NLC detections happened at times when the wave activity was low, i.e. times with no significant spectral power.

### 4.3   Start of the NLC season

Given that there are summer seasons with just a single NLC detection, using the term 'NLC season' in these cases may be deceptive. However, we decided to stick with this term since it is used in many studies. At any rate, our data show that the

first NLC detection of a summer season is highly variable, ranging from -23 days from solstice to 38 days past solstice.



**Table 3.** The meridional wind $v$ measured by SAAMER at 84 km altitude for NLC events. $v_6$ ($v_{12}$, $v_{24}$) denotes the mean meridional wind during a period of time of 6 h (12 h, 24 h) prior to the time of maximum NLC brightness. Significant tidal and planetary wave (PW) periods of $v$ are taken from the analysis presented in Fig. 6.

| Date | $v$ | $v_6$ | $v_{12}$ | $v_{24}$ (m/s) | tidal period (h) | PW period (d) |
|---|---|---|---|---|---|---|
| 10 Jan 2018 | 5 | 29 | 20 | 9 | - | - |
| 18 Jan 2018 | 30 | 26 | 24 | 23 | 16 | - |
| 24 Dec 2018 | 42 | 54 | 30 | 21 | - | - |
| 25 Dec 2018 | - | 29 | 22 | 19 | 6 | - |
| 4 Jan 2019 | 31 | 4 | 36 | 13 | - | 1.6 |
| 9 Jan 2019 | -10 | 4 | 6 | 15 | - | - |
| 29 Nov 2019 | 11 | 34 | 25 | 23 | - | - |
| 26 Dec 2019 | -7 | 10 | 7 | 10 | 7 | - |
| 29 Dec 2019 | 19 | 27 | 17 | 8 | - | - |
| 31 Dec 2019 | 1 | 24 | 26 | 24 | - | - |
| 3 Jan 2021 | 13 | 0 | 11 | 15 | 9 | - |
| 24 Jan 2022 | 10 | 19 | 25 | 22 | - | - |

These late first NLC occurrences are much later and seemingly uncorrelated with the onsets of the southern hemisphere polar mesospheric cloud (another term for noctilucent clouds used for observations from space) season as determined from two satellite experiments with values between -36 and -30 days from solstice (DeLand and Gorkavyi, 2021, their Fig. 10, 2017–2020, south of 60°S).

Karlsson et al. (2011) have shown that the timing of the breakdown of the southern hemisphere polar vortex influences the onset of the southern hemisphere NLC season. Fig. 7 shows zonal mean zonal winds at 0 UT and 60°S from ERA5 at 10 hPa. The Antarctic minor sudden stratospheric warming in September 2019 caused a reduction in wind speed that is visible at around -95 days from solstice (green curve in Fig. 7). The warming event weakened the polar vortex, resulting in an earlier breakdown. It is evident from Fig. 4b that temperatures in the NLC altitude region were low in 2019 (green curve) compared 250 to other years. The lower mesospheric temperature due to the stratospheric warming and thus the earlier start of summer has likely contributed to the early detection of NLC that occurred on 29 Nov 2019. This event is the so far only detection of NLC by CORAL at Rio Grande before summer solstice.

## 4.4 Occurrence rate

The middle of the season at Rio Grande is found on 2 Jan (12 days from solstice, median of column 2 of Table. 1. We 255 define the occurrence rate as the accumulated time of NLC detections divided by the accumulated measurement time between





**Figure 6.** Significant spectral power of the meridional wind at 84 km altitude, tested against red noise with 95 % confidence level, between 6 h and 10 d period. Hourly lidar measurements are indicated by dotted vertical lines, and such with NLC detected by solid lines. Significant matches are listed in Table 3.

27 November and 31 January of each season. These dates were chosen to include our earliest and latest observation dates, and are very similar to the dates used by Gerding et al. (2013a) (that would be 1 December to 4 February in the southern hemisphere). Measurement nights and hours per season are listed in Tab. 4. NLC were detected in one to six nights per season, with occurrence rates ranging between 2% and 12%. Considering all seasons, in total 523.7 h with lidar observations, of which
33.8 h are with NLC detections, results in an occurrence rate of 6%. This compares well to the value of 6% from 21 h with NLC detections in 2005/2006 at Davis (68.6°S) in East Antarctica found by Klekociuk et al. (2008). Chu et al. (2006) derived higher values with a mean of 27.9% from three seasons of NLC observations (2002/2003 to 2004/2005, 128 h with NLC) at Rothera



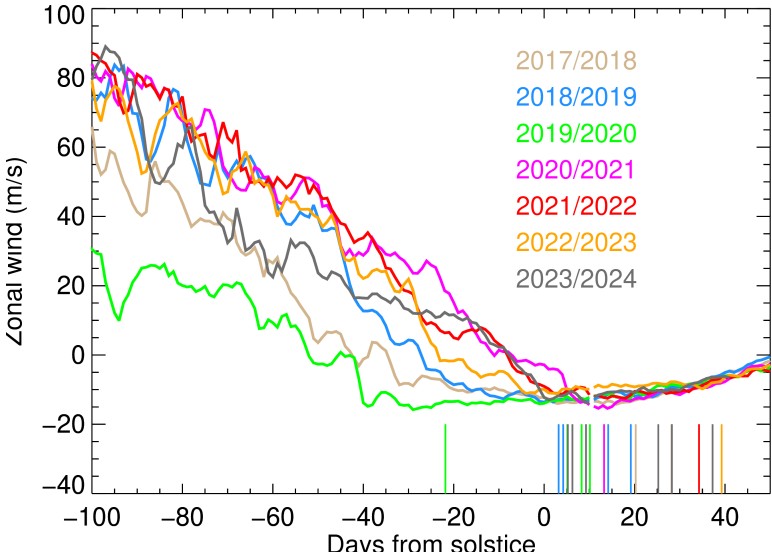

**Figure 7.** ERA5 zonal mean zonal wind at 0 UT, 10 hPa and -60° latitude. Solstice refers to 21 December. NLC occurrences observed by CORAL are indicated by vertical lines at the bottom.

**Table 4.** Measurement statistics and NLC occurrence rate between 27 Nov and 31 Jan. Mean Ly-alpha is taken from Machol et al. (2019).

| Season | Measurement (nights) | Measurement (h) | NLC (nights) | NLC (h) | Occurrence rate (%) | Ly-alpha ($10^{-3}\text{W/m}^2$) |
|--------|--------|--------|--------|--------|--------|--------|
| 2017/2018 | 20 | 37.8 | 2 | 4.0 | 11 | 6.3 |
| 2018/2019 | 39 | 85.8 | 4 | 6.6 | 8 | 6.1 |
| 2019/2020 | 38 | 74.1 | 4 | 5.4 | 7 | 6.1 |
| 2020/2021 | 28 | 51.8 | 1 | 1.7 | 3 | 6.6 |
| 2021/2022 | 42 | 98.8 | 1 | 3.8 | 4 | 7.0 |
| 2022/2023 | 41 | 83.1 | 1 | 1.5 | 2 | 8.2 |
| 2023/2024 | 50 | 92.3 | 6 | 10.8 | 12 | 8.6 |
| all | | 523.7 | 19 | 33.8 | 6 | |

(67.5°S), which is located at a comparable longitude but higher latitude than Rio Grande. Their values were however calculated from a shorter core season and vary significantly over the three seasons (as in our case) since dominated by singular events in individual years. Indications of strong inter-annual variability also come from Gerding et al. (2013a), who report a variability between 0 and 19% from 15 years of lidar observations (1997–2011) at Kühlungsborn (54°N). Surprisingly, the 15-year mean is 6%, the same value we find for Rio Grande. From satellite measurements in the two seasons 2017/2018 and 2018/2019




DeLand and Thomas (2019, their Fig. 4) derived an occurrence rate in the 50°–64° latitude band of the southern hemisphere of just 0.5%. This rate is a factor of four smaller compared to about 2% for the same band in the northern hemisphere. Given

the obviously large variability, it is hard to reliably compare occurrence rates at different sites, but looking at the available data, the occurrence rate at Rio Grande seems to be higher than what would be expected in terms of ambient temperature, latitude and inter-hemispheric differences.

A likely source of inter-annual variability is the variation of the solar flux with the solar cycle. Solar Lyman alpha radiation photodissociates water vapour, resulting in fewer NLC during solar maximum (Garcia, 1989). DeLand et al. (2003) found an

anti-correlation with no phase lag in the southern hemisphere from a satellite record spanning two solar cycles. Our observations are limited to seven years within solar cycles 24 and 25. As evident from Tab. 4, the occurrence rate is high during the first three seasons (solar mininum) and seemingly low thereafter when the composite solar Ly-$\alpha$ index increases. The last season of 2023/2024 at the peak of the solar maximum however showed unexpectedly high occurrence rates and durations. In fact, this is the season with the most NLC events (six nights) so far. We conclude that for our observations at a mid-latitude site in the

southern hemisphere solar activity with solar Lyman alpha as proxy is not a main driver for the observed variability.

An additional source of water vapour in the upper mesosphere and lower thermosphere that might trigger bright NLC especially at mid-latitudes is rocket engine exhaust (Stevens et al., 2012; Siskind et al., 2013; Stevens et al., 2022). The first evidence was presented by Stevens et al. (2005a) who found increased NLC occurrence rates and larger ice mass in the northern hemisphere up to 8 days after a space shuttle launch. Soon after, the thermospheric transport of a space shuttle exhaust

plume containing water vapour and iron into the southern hemisphere was investigated using lidar and satellite measurements. Stevens et al. (2005b) showed that the exhaust plume reached the Antarctic Peninsula within three days. Since then, the number of rocket launches has dramatically increased. We looked at launch dates of rockets launched from major space centers that may potentially be linked to NLC observed above Rio Grande. These centers are the Guiana Space Center in Kourou, French Guiana (6700 km to the north), the Kennedy Space Center and Cape Caneveral Space Force Station (9300 km to the north),

and New Zealands' Mahia peninsula which is 8000 km west of Rio Grande. Exhaust from the Electron rockets launched at Mahia peninsula may be transported to Rio Grande by the eastward winds above 85 km altitude as shown by Hindley et al. (2021), however these smaller rockets inject only a small amount of water vapour into the lower thermosphere. Close temporal proximity of Electron rocket launches and NLC observations by lidar at Rio Grande are found for 16 December 2018 (8 days prior to a NLC event), 24 January 2023 (5 days prior) and 15 December 2023 (11 days prior). Soyuz launches from Kourou

took place on 19 December 2018, 18 December 2019, 29 December 2020, and 21 January 2022, that is 5, 8, 5, and 3 days prior to NLC observations at Rio Grande. For almost all NLC events we find Falcon 9 launches from Kennedy Space Center within 3 to 12 days prior to NLC observations. Given the large number of possibilities, better modelling of the transport of the rocket exhaust by wind, taking into accounts the strong tidal winds in the lower thermosphere, is needed to confirm a potential direct effect of rocket exhaust on the occurrence of NLC. At this point, the temporal proximity and large number of rocket launches

make space traffic a potential factor contributing to the unexpectedly large number of NLC events observed above Rio Grande.

There was also major natural event which potentially contributed to the large number of NLC observations in the 2023/2024 season. The eruption of the Hunga Tonga undersea volcano in January 2022 injected large amounts of water vapour into



the stratosphere, which then reached the mesosphere several months later (Nedoluha et al., 2023). The eruption also induced changes in circulation patterns of the middle atmosphere with effects on mesospheric temperature (Yu et al., 2023). However, detailed model studies and observations of water vapour in particular are necessary to investigate these effects. Such work is beyond the scope of this paper.

## 5 Conclusions

We reported on NLC observations by lidar at Rio Grande, Tierra del Fuego, which is located at mid-latitudes in the southern hemisphere (53.8°S). The 19 events detected in seven summer seasons represent the first NLC observations by lidar in the southern hemisphere outside of Antarctica. The occurrence rate of about 6% is unexpectedly high and comparable to measurements at a conjugate latitude in the northern hemisphere. The mean altitude (83.3 km) is higher than in the northern hemisphere. We observed several events of high brightness that lasted several hours. The local time dependence of NLC observations reveals the influence of a pronounced semi-diurnal tidal wind that transports ice particles from Antarctic latitudes to the north towards Rio Grande. Northward transport seems to be crucial because, as confirmed by our measurements, the mean temperature at the height of NLC above Rio Grande is well above the frost point and thus effectively prevents local formation of ice particles. Our lidar temperature measurements agree with previous assessments of inter-hemispheric differences of the local thermal background. Another factor might be increased amounts of water vapour by man-made or natural causes. Whether there are direct links to orbital rocket launches should be studied with modelling in the future. Here, we conclude that unexpectedly bright NLC occured above Rio Grande. These NLC dislays must be visible by naked eye possibly up to hundreds of kilometers to the north. That there are very few visual reports from South America, unlike in the northern hemisphere, may be attributed to populated areas being few and sparse. Whether the appearance of bright NLC above Tierra del Fuego is anomalous with respect to longitude, similar to the winter stratospheric GW hotspot caused by the Andes, is unknown.

*Data availability.* Lidar data are available from the HALO database https://halo-db.pa.op.dlr.de/mission/111. ERA5 reanalysis data is available from https://cds.climate.copernicus.eu. SABER data is available from https://saber.gats-inc.com/. Lyman alpha index is accessible via the LASP Interactive Solar Irradiance Datacenter (LISIRD, https://lasp.colorado.edu/lisird/).

*Author contributions.* BK and NK built and operated the CORAL lidar. BK and NK performed the data analysis and wrote the manuscript. GL and DJ provided meteor radar data. GB provided camera images and projections. JH supported lidar operations in Rio Grande. All authors revised the manuscript.

*Competing interests.* The authors declare no competing interests.



330  *Acknowledgements.* NK and BK thank Alejandro de la Torre for his support in bringing CORAL to Rio Grande, and Robert Reichert for help with the initial installation and servicing of the instrument. GB and NK acknowledge the work of the members of the German forum Arbeitskreis Meteore (https://www.meteoros.de/), Simon Herbst in particular, who screen and discuss sightings of NLC from all available online sources including the IAP camera network, and who discovered several of the NLC events mentioned here in the real-time camera images. GB thanks Michael Priester for screening the IAP camera images. NK thanks Isabell Krisch for comments.



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
