# Peer review of "Lidar measurements of noctilucent clouds at Rio Grande, Tierra del Fuego, Argentina"

_EGUsphere, 2024_

## Referee Comment (RC2)

A review on the paper "Lidar measurements of noctilucent clouds at Rio Grande, Tierra del Fuego, Argentina" by Natalie Kaifler, Bernd Kaifler, Markus Rapp, Guiping Liu, Diego Janches, Gerd Baumgarten and Jose-Luis Hormaechea.

**General comments**:

In the present paper, the authors investigate noctilucent clouds (NLC) in the southern hemisphere at mid-latitude sites in southern Argentina. The authors have been using lidar soundings since 2017 and optical camera observations since 2019, enabling comparisons between the vertical lidar soundings and NLC spatial imaging in a common volume of the mesopause. The authors have found in total 19 NLC events, at an average height of 83.3 km, an average NLC occurrence rate of 6% and a NLC maximum in the morning hours. The ambient temperature above the lidar site was, on average, too high to support local NLC ice particle formation. The authors explain this by the northward transport of NLC. Another explanation is a possible influence of increasing space traffic producing enhanced amount of water vapor in the upper mesosphere.

I have found the present paper to be very interesting to the atmospheric community. I recommend the present paper for publication after minor revisions which are outlined below.

**Specific comments:**

Page 1, L.3: "At northern hemisphere mid-latitudes, the occurrence of NLC seems to increase with time."

This sentence should be removed from the abstract since it is debatable in the literature and it is beyond the scope of the present manuscript.

Page 1, L.16-17: "Noctilucent clouds (NLC) were discovered at northern-hemispheric mid-latitudes by visual observation of the horizon in twilight conditions (Backhouse, 1885; Jesse, 1885; Leslie, 1885)."

Please add here the paper by Tseraskii (1887) who observed, photographed and estimated the NLC altitude for the first time already in June 1885.

Page 2, L.25-28: " In recent decades, the number of observations in the northern hemisphere appeared to increase, and efforts were made to uncover the origins of mid-latitude NLC and study their possible relation to climate change (von Cossart et al., 1996; Nielsen et al.,2011; Hultgren et al., 2011; Gerding et al., 2013a; Russell III et al., 2014; Hervig et al., 2016)."

This is not the whole story of the topic regarding "the number of in the northern hemisphere appeared to increase…". The authors traditionally highlight one side of this topic only, and traditionally forgetting another side of this problem. There is a series of scientific publications clearly demonstrating a slight positive **statistically insignificant trend** or **about zero trend** in the NLC occurrence number and NLC brightness at middle latitudes (Dalin et al., 2020, Dubietis et al., 2010; Kirkwood et al., 2008; Kirkwood and Stebel, 2003; Pertsev et al., 2014; Zalcik et al., 2014, 2016). If the authors really want to highlight this topic then the authors should illuminate another side of this problem on about zero trend in the NLC occurrence at mid-latitudes as well (see Dalin et al., 2020 and references therein). Otherwise, this topic should be removed since it is beyond the scope of the present manuscript.

Page 2, L.40-42: "Prior to the deployment, no sightings of NLC north of 54°S have ever been reported from this or any other longitude in the southern hemisphere."

This sentence sounds strange. It is well-known that a big network of NLC observing stations was established in the southern hemisphere (between 45°S and 90°S) in the 1960s.

In particular, there were registered and photographed several NLC displays from Punta Arenas 53.1°S, 71.0°W (Chile). Besides, one can read the following statement from Fogle and Haurwitz (1966): "The brightest and most widespread displays observed on the expedition occurred during the period January 1-4 with the best display taking place on the night of January 3 (this display was also observed by personnel at the weather stations at **Port Stanley (51.7°S, 57.9°W) in the Falkland Islands**…". Thus, this is certainly "north of 54°S". Please see publications by Fogle (1964, 1965), Fogle and Haurwitz (1966) on NLC observations in the SH.

Page 6, L.128-129:  "… the NLC layer is perturbed at periods below the Brunt-Vaisala period."
Some comment is needed here to explain how it is possible to observe periods in an NLC layer below the Brunt-Vaisala period.

Page 7, L.146-147: "In fact, NLC can be detected in camera images at locations close to the lidar beam and yet remain unvisible to the lidar."
Some comment is needed here to explain why NLC might be unvisible to the lidar, having NLC in images close to the position of the lidar beam.

Page 8-9, L.168-169: "Evidence for the very same small-scale dynamics are also found throughout the lidar soundings in the form of very short-period (around 1 min) modulations of the NLC layer with vertical displacements of few hundred meters."
In my opinion, this is a very interesting result which is better to demonstrate it in a figure. Please add a figure showing small-scale short-period (1 min) dynamics of the lidar sounding for this case.

Page 12, L.211: "The peak in brightness between 5–7:30 UT…"
Please add LT here as well.

Page 12, L.226: " Spectral power for periods between 6 h and 16 d are presented in Fig. 6"
In Fig. 6, one can see the period scale until 10 days. Where can I find periods of more than 10 days?

Page 13, L.232-233: "(Merkel et al., 2003) and (Merkel et al., 2008) detected a pattern in noctilucent cloud brightness related to the 5-day planetary wave from satellite measurements."
Here it is worth mentioning two papers by Dalin et al. (2008; 2011) which clearly demonstrated the influence of 2- and 5-day planetary waves on NLC activity.

Page 17, L.281-282: "An additional source of water vapour in the upper mesosphere and lower thermosphere that might trigger bright NLC especially at mid-latitudes is rocket engine exhaust (Stevens et al., 2012; Siskind et al., 2013; Stevens et al., 2022)."
Here it is worth mentioning the paper by Dalin et al. (2013) which clearly demonstrated the direct formation of NLC in the rocket exhaust trail.

**Additional references:**
Dalin, P., Perminov, V., Pertsev, N., and Romejko, V.: Updated long-term trends in mesopause temperature, airglow emissions, and noctilucent clouds. Journal of Geophysical Research-Atmospheres, 125, e2019JD030814, https://doi.org/10.1029/2019JD030814, 2020.

Dalin, P., Perminov, V., Pertsev, N., Dubietis, A., Zadorozhny, A., Smirnov, A., Mezentsev, A., Frandsen, S., Grönne, J., Hansen, O., Andersen, H., McEachran, I., McEwan, T., Rowlands, J., Meyerdierks, H., Zalcik, M., Connors, M., Schofield, I., Veselovsky, I.: Optical studies of rocket exhaust trails and artificial noctilucent clouds produced by Soyuz rocket launches, JGR-Atmospheres, 118, 14, 7850-7863, doi:10.1002/jgrd.50549, 2013.

Dalin, P., N. Pertsev, A. Dubietis, M. Zalcik, A. Zadorozhny, M. Connors, I. Schofield, T. McEwan, I. McEachran, S. Frandsen, O. Hansen, H. Andersen, V. Sukhodoev, V. Perminov, R. Balčiunas, V. Romejko: A comparison between ground-based observations of noctilucent clouds and Aura satellite data. J. Atmos. Solar-Terr. Phys., 73, 14-15, 2097-2109, doi:10.1016/j.jastp.2011.01.020, 2011.

Dalin, P., N. Pertsev, A. Zadorozhny, M. Connors, I. Schofield, I. Shelton, M. Zalcik, T. McEwan, I. McEachran, S. Frandsen, O. Hansen, H. Andersen, V. Sukhodoev, V. Perminov, V. Romejko: Ground-based observations of noctilucent clouds with a northern hemisphere network of automatic digital cameras. J. Atmos. Solar-Terr. Phys., 70, 11-12, 1460-1472, doi:10.1016/j.jastp.2008.04.018, 2008.

Dubietis, A., Dalin, P., Balciunas, R., & Cernis, K.: Observations of noctilucent clouds from Lithuania. Journal of Atmospheric and Solar - Terrestrial Physics, 72(14-15), 1090–1099, https://doi.org/10.1016/j.jastp.2010.07.004, 2010.

Fogle, B.: Noctilucent clouds in the southern hemisphere. Nature, 14, 204, 1964.

Fogle, B.: Noctilucent clouds over Punta Arenas, Chile. Nature, 66, 207, 1965.

Fogle, B., and Haurwitz, B.: Noctilucent clouds. Space Science Reviews, 6, 3, 279-340, 1966.

Kirkwood, S., Dalin, P., and Réchou, A.: Noctilucent clouds observed from the UK and Denmark—Trends and variations over 43 years. Annales Geophysicae, 26, 1243–1254, 2008.

Kirkwood, S., and Stebel, K.: Influence of planetary waves on noctilucent cloud occurrence over NW Europe. Journal of Geophysical Research, 108(D8), 8440. https://doi.org/10.1029/2002JD002356, 2003.

Pertsev, N., Dalin, P., Perminov, V., Romejko, V., Dubietis, A., Balčiunas, R., et al.: Noctilucent clouds observed from the ground: sensitivity to mesospheric parameters and long-term time series. Earth, Planets and Space, 66(1), 1–9, https://doi.org/10.1186/1880-5981-66-98, 2014.

Tseraskii, V. K.: Astronomichesky fotometr i ego prilozhenia (Astronomical photometer and its applications). Doctoral Dissertation, Mathematical Proceedings, XIII, Section 21, 626–631, 1887 (in Russian).

Zalcik, M. S., Lohvinenko, T. W., Dalin, P., and Denig, W. F.: North American noctilucent cloud observations in 1964–77 and 1988–2014: Analysis and comparisons. Journal of the Royal Astronomical Society of Canada, 110(1), 8–15, 2016.

Zalcik, M. S., Noble, M. P., Dalin, P., Robinson, M., Boyer, D., Dzik, Z., et al.: In search of trends in noctilucent cloud incidence from the La Ronge flight service station (55°N 105°W). Journal of the Royal Astronomical Society of Canada, 108(4), 148–155, 2014.

---

## Author Comment (AC1)

**Response (black) to the reviewer's comments (blue italics) by the authors:**

We thank the reviewer for careful reading and especially pointing to present and past works on the topic.

*Page 1, L.3: "At northern hemisphere mid-latitudes, the occurrence of NLC seems to increase with time." This sentence should be removed from the abstract since it is debatable in the literature and it is beyond the scope of the present manuscript.*

The sentence was deleted as suggested.

*Page 1, L.16-17: "Noctilucent clouds (NLC) were discovered at northern-hemispheric midlatitudes by visual observation of the horizon in twilight conditions (Backhouse, 1885; Jesse, 1885; Leslie, 1885)." Please add here the paper by Tseraskii (1887) who observed, photographed and estimated the NLC altitude for the first time already in June 1885.*

The reference was added as suggested, thank you.

*Page 2, L.25-28: " In recent decades, the number of observations in the northern hemisphere appeared to increase, and efforts were made to uncover the origins of mid-latitude NLC and study their possible relation to climate change (von Cossart et al., 1996; Nielsen et al.,2011; Hultgren et al., 2011; Gerding et al., 2013a; Russell III et al., 2014; Hervig et al., 2016)."*
*This is not the whole story of the topic regarding "the number of in the northern hemisphere appeared to increase...". The authors traditionally highlight one side of this topic only, and traditionally forgetting another side of this problem. There is a series of scientific publications clearly demonstrating a slight positive statistically insignificant trend or about zero trend in the NLC occurrence number and NLC brightness at middle latitudes (Dalin et al., 2020, Dubietis et al., 2010; Kirkwood et al., 2008; Kirkwood and Stebel, 2003; Pertsev et al., 2014; Zalcik et al., 2014, 2016). If the authors really want to highlight this topic then the authors should illuminate another side of this problem on about zero trend in the NLC occurrence at mid-latitudes as well (see Dalin et al., 2020 and references therein). Otherwise, this topic should be removed since it is beyond the scope of the present manuscript.*

We admit that the introduction was kept too short, although some references dealing with this complex topic were given. As some aspects like solar cycle and planetary wave effects are discussed later in the manuscript, we chose to extend the introduction and included all suggested citations (thank you) as well as the two noted by the other reviewer. The text was changed to:

"The formation of NLC sensitively depends on temperature and the available water vapour \citep{Hervig2016}, making them a potential indicator of climate change \citep{Thomas1996}. Establishing long-term trends however requires careful analysis of datasets and correction of varying responses of NLC to the solar cycle and planetary waves \citep{DeLand2002,Kirkwood2003,Gerding2012,Russell2013,Hervig2019}. Although with the rise of digital cameras amateur reports from Europe and North America are now numerous, studies of long-term ground-based and satellite records found no general long-term trend \citep{Kirkwood2008,Dubietis2010,Pertsev2014,Zalcik2014,Zalcik2016,Dalin2020}."

*Page 2, L.40-42: "Prior to the deployment, no sightings of NLC north of 54°S have ever been reported from this or any other longitude in the southern hemisphere." This sentence sounds strange. It is well-known that a big network of NLC observing stations was established in the southern hemisphere (between 45°S and 90°S) in the 1960s. 2 In particular, there were registered and photographed several NLC displays from Punta Arenas 53.1°S, 71.0°W (Chile). Besides, one can read the following statement from Fogle and Haurwitz (1966): "The brightest and most widespread displays observed on*

*the expedition occurred during the period January 1-4 with the best display taking place on the night of January 3 (this display was also observed by personnel at the weather stations at Port Stanley (51.7°S, 57.9°W) in the Falkland Islands…". Thus, this is certainly "north of 54°S". Please see publications by Fogle (1964, 1965), Fogle and Haurwitz (1966) on NLC observations in the SH.*

That's true. The observations by Benson Fogle in the 1960s were mentioned in the first paragraph. He himself doubted two historic reports from Punta Arenas. He observed one display in Januar 1965. In the austral summer of 1965/1966, a dedicated observation campaign including aircraft resulted in nine observations between 25 Dec and 20 Jan, two of them bright. That matches quite well actually with our observations. These are the only reported results from the network shown in Fig. 5 of Fogle and Haurwitz (1966). Until a photograph from New Zealand from 1 Dec 2019, I found no further reports of visual NLC from outside the Antarctic continent. This is likely for lack of observers. The sentence was modified and the references to Fogle added more prominently to the introduction.

"Ground-based camera network were very successfull in systematically observing the skies for the occurrence of noctilucent clouds (Witt1962, Fogle and Haurwitz, 1966, Dalin et al., 2009, Dubietis et al., 2010)."

"Historical observations from the southern hemisphere are rare and limited to the reports by Jesse (1889) and (Fogle, 1965; Fogle and Haurwitz, 1966) from Punta Arenas.."

"Prior to the deployment, no recent NLC observations were known from this latitude."

*Page 6, L.128-129: "… the NLC layer is perturbed at periods below the Brunt-Vaisala period." Some comment is needed here to explain how it is possible to observe periods in an NLC layer below the Brunt-Vaisala period.*

Instabilities such as Kelvin-Helmholtz instabilities induced by breaking waves could exhibit such short periods. The process is mentioned in the paragraph and the sentence modified to "at periods below the Brunt-Vaisala period, indicative of instabilities."

*Page 7, L.146-147: "In fact, NLC can be detected in camera images at locations close to the lidar beam and yet remain unvisible to the lidar." Some comment is needed here to explain why NLC might be unvisible to the lidar, having NLC in images close to the position of the lidar beam.*

That might occur due to local variability in the NLC layer, that is sometimes patchy or has holes. The lidar only has a very small field of view and thus footprint at the NLC layer on the order of 1 m, which makes it very sensitive to local variations of NLC brightness. Thus seeing NLC in the general direction is no guarantee that it can be detected in that exact spot the lidar points to.

The sentence was changed to: "If the NLC layer is very patchy, it might be possible that although NLC is visible at locations close to lidar beam, it is not detected in the small spot observed by the lidar."

*Page 8-9, L.168-169: "Evidence for the very same small-scale dynamics are also found throughout the lidar soundings in the form of very short-period (around 1 min) modulations of the NLC layer with vertical displacements of few hundred meters." In my opinion, this is a very interesting result which is better to demonstrate it in a figure. Please add a figure showing small-scale short-period (1 min) dynamics of the lidar sounding for this case.*

Here is a zoomed-in version of Fig. 2e showing very short scale motions if rather low amplitude that are superposed on the brighter layer. We add this as suggested as sub-figure to Fig. 3. Another nice example of also about 1-min oscillations occurs on 10 Jan 2018 (Fig. 2a), enlarged also below, showing that this is not an uncommon feature.

[Figure]

*Page 12, L.211: "The peak in brightness between 5–7:30 UT…" Please add LT here as well.*

changed to 5--7:30~UT (2--4:30~LT) as Argentina time zone is UTC-3

*Page 12, L.226: " Spectral power for periods between 6 h and 16 d are presented in Fig. 6" In Fig. 6, one can see the period scale until 10 days. Where can I find periods of more than 10 days?*

Thank you for pointing this discrepancy out. Indeed I modified the plot axis because no significant periods of more than 10 days were detected. The time series might also be too short to reliably detect 16-day waves by this technique. We adjusted the figure axis.

*Page 13, L.232-233: "(Merkel et al., 2003) and (Merkel et al., 2008) detected a pattern in noctilucent cloud brightness related to the 5-day planetary wave from satellite measurements." Here it is worth mentioning two papers by Dalin et al. (2008; 2011) which clearly demonstrated the influence of 2- and 5-day planetary waves on NLC activity.*

Thank you, the references were added in this paragraph and in the introduction.

*Page 17, L.281-282: "An additional source of water vapour in the upper mesosphere and lower thermosphere that might trigger bright NLC especially at mid-latitudes is rocket engine exhaust (Stevens et al., 2012; Siskind et al., 2013; Stevens et al., 2022)." Here it is worth mentioning the paper by Dalin et al. (2013) which clearly demonstrated the direct formation of NLC in the rocket exhaust trail.*

The reference was added.

---

## Author Comment (AC2)

**Response (black) to the reviewer's comments (blue italics) by the authors:**

We thank the reviewer for his input and inspiring questions.

*Page 9, lines 169-171: Now that you have demonstrated the ability to visually identify NLCs at Southern Hemisphere mid-latitudes, have you considered trying to recruit volunteer observers at appropriate locations in Argentina to supplement your measurements?*

Yes, also thanks to your question, we felt encouraged that this is indeed worth it. One of us is in contact with astro-photographers and astronomers that are active in Argentina and will support them in making suitable observations. We will also contact spaceweather.com (which had a story on one of the events observed from Rio Grande before) to post a request with the start of the next season in November. It occurred to me that it would also be beneficial to create and distribute teaching material for primary school students on the topic. I have some ideas for hands-on physics experiments that I will pursue.

*Page 12, lines 205-208: You have noted that your NLC observations are influenced by special conditions (e.g. gravity waves, meridional transport). Are there reasons to believe that these conditions would consistently produce higher altitudes and larger vertical extent for NLC?*

That is a very good question. I (nk) think it might be possible. In a very quiet, natural environment, NLC particles can reach low altitudes and form thin, unperturbed layers. Special conditions will likely increase variability, that is if a specific duct is at a certain altitude, NLC will form there. It might be a different altitude in the next case.  Strong wave activity will lead to even a thin layer populating a wider vertical range. That might mean that even more statistics is needed to arrive at a reliable mean value. But with this picture of quietly sedimenting, growing ice particles that sublimate almost instantaneously at the lower boundary, it is plausible that such a setting will result in the lowest mean altitudes, in contrast to a strongly perturbed environment.

*Page 12, lines 210-212: Local time dependence is certainly present in Northern Hemisphere lidar NLC data, with peak occurrence frequency and brightness in the early morning [e.g. Fiedler et al., 2017, J. Atmos. Solar-Terr. Phys. 162, 79-89].*

We added "Considerable local time variations with peak occurrence frequency and brightness in the early morning are known from northern hemisphere observations \citep{Fiedler2017}."

*Page 17, lines 279-280: You may wish to note that the response of NLCs to solar variations has been significantly reduced since the early 2000s, as discussed in some recent papers [e.g. Hervig et al., 2019, Geophys. Res. Lett. 46, 10,132-10,139; Vellalassery et al., 2023, Ann. Geophys. 41, 289-300].*

The text was extended: "A potential source of inter-annual variability is the variation of the solar flux with the solar cycle. Solar Lyman alpha radiation photodissociates water vapour, resulting in fewer NLC during solar maximum \citep{Garcia1989}. \citet{DeLand2002} found an anti-correlation with no phase lag in the southern hemisphere from a satellite record spanning two solar cycles. After 2020, however, satellite and model results suggest a significantly reduced response of noctilucent clouds to solar variations \citep{Hervig2019,Vellalassery2023}."

*Page 17, lines 297-299: Previous studies do show the complex nature of possible attribution of NLC formation (or enhancement) to rocket exhaust. However, given the unfavorable normal background conditions at this location, episodic water vapor enhancement is certainly a viable option, and may be worth investigation for selected cases.*

Thank you for your comment. It will be interesting to continue observations into the future, and even if the interplays are complex, maybe a trend will eventually crystallize. The pollution of the MLT region by the exponentially increasing space traffic with both exhaust and debris might result in numerous effects not limited to NLC in the future. I think this development demands monitoring by scientists.

*Page 18, line 319: "dislays" should be "displays".*

Fixed, thank you.